# Diesel soot photooxidation enhances the heterogeneous formation of $H_2SO_4$

Peng Zhang[1], Tianzeng Chen[1,2], Qingxin Ma [1,2,3] ✉, Biwu Chu [1,2,3], Yonghong Wang[1,2], Yujing Mu [1,2,3], Yunbo Yu[1,2,3] & Hong He [1,2,3] ✉

Both field observation and experimental simulation have implied that black carbon or soot plays a remarkable role in the catalytic oxidation of $SO_2$ for the formation of atmospheric sulfate. However, the catalytic mechanism remains ambiguous, especially that under light irradiation. Here we systematically investigate the heterogeneous conversion of $SO_2$ on diesel soot or black carbon (DBC) under light irradiation. The experimental results show that the presence of DBC under light irradiation can significantly promote the heterogeneous conversion of $SO_2$ to $H_2SO_4$, mainly through the heterogeneous reaction between $SO_2$ and photo-induced OH radicals. The detected photochemical behaviors on DBC suggest that OH radical formation is closely related to the abstraction and transfer of electrons in DBC and the formation of reactive superoxide radical ($\cdot O_2^-$) as an intermediate. Our results extend the known sources of atmospheric $H_2SO_4$ and provide insight into the internal photochemical oxidation mechanism of $SO_2$ on DBC.

The rapid increase in vehicle numbers has resulted in the emission of large quantities of black carbon (BC) into the lower atmosphere[1–4]. BC particles in vehicle exhaust are mainly formed by incomplete combustion of hydrocarbon fuels[5–7]. In particular, in northern China, BC mass concentrations of up to $20\,\mu g\,m^{-3}$ (nearly 10% of the total particulate matter) have been observed during haze episodes[8]. High loading of BC can increase atmospheric stability through the formation of a temperature inversion, which will feed back to the development of extreme haze via suppressing air pollutant dispersion[9]. Moreover, BC aerosol can influence climate by directly absorbing solar radiation and affecting cloud formation and surface albedo through deposition on snow and ice[10–15].

Recent works proved that the interactions between BC and other inorganic species can enhance the atmospheric oxidation capacity and contribute to the formation of complex air pollution[16,17]. For instance, gaseous nitrous acid (HONO) is an important precursor of hydroxyl radical (OH) in the troposphere. Numerous studies have shown that the heterogeneous reduction of $NO_2$ on the BC surface is an important HONO source[18–22]. Moreover, sulfate is the fastest-forming species and rapidly becomes the main component of secondary aerosols during the evolution of haze[23–28]. Recent laboratory simulation and theoretical calculation works have indicated that soot can act as a catalyst to promote the heterogeneous oxidation of $SO_2$ to sulfate under dark conditions[29–31]. Chamber experiments also proved that the catalytic role of soot in sulfate formation can be further amplified by reducing $NO_2$ to HONO in the presence of both $NO_2$ and $NH_3$ under dark conditions[32]. A recent field measurement in urban Beijing conducted by Yao et al. indicated that the catalytic oxidation of $SO_2$ on traffic-related soot can induce the formation of gas-phase $SO_3$ in the early morning[33]. Therefore, the heterogeneous chemistry involving soot has recently attracted increasing attention in the field of atmospheric chemistry.

Relative to its role under dark conditions, some recent studies reported that both elemental carbon (EC) and organic carbon (OC) in BC exhibited conspicuous photo-reactivity due to their strong light absorption capability under illumination[19,34–37]. The EC-initiated photo-oxidation of OC was found to proceed through radical reactions initiated by electron transfer, and the absorption of light by OC-induced direct photoaging by energy transfer. Recent observational evidence suggested that photochemical reactions on soot particles may contribute to the production of atmospheric sulfate during the daytime[38].

[1]State Key Joint Laboratory of Environment Simulation and Pollution Control, Research Center for Eco-Environmental Sciences, Chinese Academy of Sciences, 100085 Beijing, China. [2]University of Chinese Academy of Sciences, 100049 Beijing, China. [3]Center for Excellence in Regional Atmospheric Environment, Institute of Urban Environment, Chinese Academy of Sciences, 361021 Xiamen, China. ✉e-mail: qxma@rcees.ac.cn; honghe@rcees.ac.cn

However, the intrinsic reaction mechanism of $SO_2$ on BC remains poorly resolved. Moreover, whether photoinduced radical chemistry can contribute to the heterogeneous conversion of $SO_2$ remains unclear.

In this work, a series of laboratory experiments are conducted to explore the underlying $SO_2$ oxidation mechanism on soot particles emitted from diesel vehicles (DBC) under light irradiation. In-situ diffuse reflectance infrared Fourier transform spectroscopy (DRIFTS) measurements and thermogravimetric analysis mass (TGA–MS) analysis are applied to characterize the formation of surface $H_2SO_4$ on DBC under light irradiation. Combined with the reactive oxygen species (ROS) analysis from electron spin resonance (ESR), it is found that OH radical resulting from the conversion of superoxide radicals is the crucial oxidant during the heterogeneous conversion of $SO_2$ to $H_2SO_4$. These results help in understanding the sources of atmospheric oxidation capacity and particulate $H_2SO_4$.

## Results

### Formation and characterization of sulfur-containing products on DBC surface

The composition and structure of DBC were firstly analyzed and characterized using in-situ DRIFTS, XPS, and XRD methods. The detailed characterization results for DBC are shown in Fig. S1. In brief, DBC was found to be amorphous and highly graphitized. Various oxygen-containing groups (such as C=O, C−OH, and C−O−C) were detected in DBC through FTIR and XPS analysis.

Heterogeneous conversion of $SO_2$ on the DBC surface was systematically investigated using in-situ DRIFTS. The main peak at 1100 cm$^{-1}$ represents the characteristic vibrations of S=O bonds in sulfur-containing products (Fig. 1a)[39]. The significant increase in peak intensity with reaction time indicated that the heterogeneous reaction between $SO_2$ and DBC resulted in the production and gradual accumulation of sulfur-containing products. As shown in Fig. 1b, c, both the presence of DBC and light irradiation can remarkably promote the heterogeneous formation of sulfur-containing species compared to reaction under dark conditions and in blank experiments. This implies that the remarkable heterogeneous conversion of $SO_2$ to sulfur-containing species should be closely related to the photo-induced catalytic role of DBC. From Fig. S2, the observation of similar in-situ DRIFTS experimental results implies that DBC photooxidation would most likely promote the heterogeneous oxidation of $SO_2$ under low-level $SO_2$ conditions (~1 ppm).

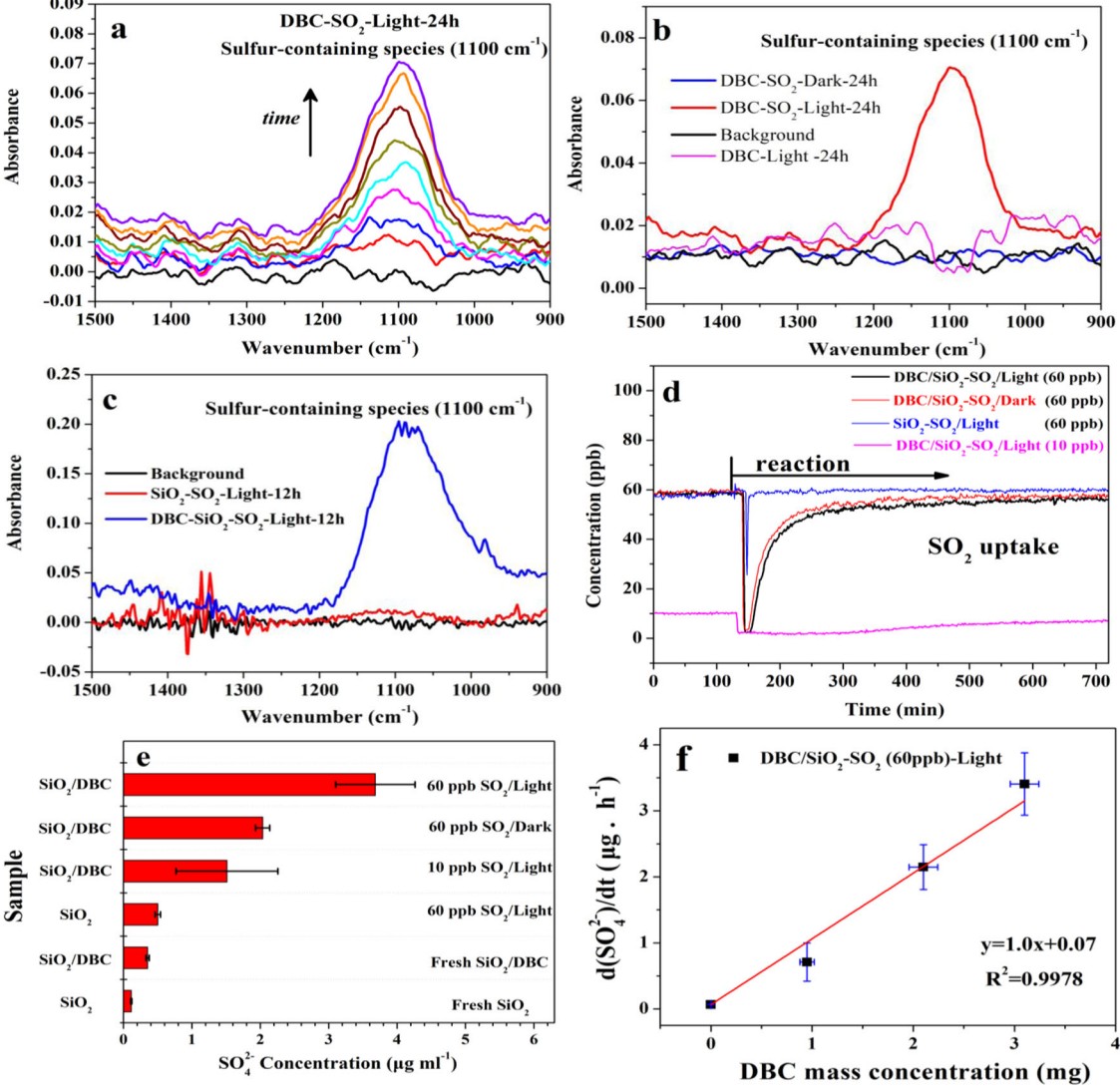

**Fig. 1 | Formation and yield of sulfur-containing products on DBC surface.** In-situ DRIFTS spectra of DBC exposed to 10 ppm $SO_2$ for 24 h under irradiation (**a**); comparison of sulfur-containing species formation under light and dark conditions (**b**); comparison of sulfur-containing species formation in the absence and presence of DBC under light irradiation (**c**); $SO_2$ uptake on DBC (0.002 g)/$SiO_2$(1 g) and $SiO_2$(1 g) under different conditions (**d**); comparison of $H_2SO_4$ concentrations under different conditions (**e**); $H_2SO_4$ formation rate as a function of DBC concentration (**f**). Error bars represent standard deviation.

Considering that typical atmospheric concentrations of $SO_2$ are at the ppb level (Fig. S3), the heterogeneous conversion of $SO_2$ (10 and 60 ppb) on the DBC surface was carried out in a tube plug flow reactor (Fig. S4). Figure 1d shows the uptake curves of 60 and 10 ppb $SO_2$ on DBC/$SiO_2$ or $SiO_2$ particles. It was found that the uptake of $SO_2$ (60 ppb) on DBC/$SiO_2$ mixtures is much greater than that on $SiO_2$. Moreover, the uptake of 10 ppb $SO_2$ on DBC is also observed to last more than 10 h under light irradiation. These results indicate that the heterogeneous uptake of $SO_2$ on DBC is significant even under conditions close to the real atmosphere. The comparison of extracted $SO_4^{2-}$ ions from different samples (Fig. 1e) further highlights the enhancing role of DBC on the heterogeneous oxidation of $SO_2$ under both dark conditions and light irradiation. The formation of $SO_4^{2-}$ under dark conditions may be due to the catalytic oxidation of $SO_2$ on the surface-active site of soot as reported in previous studies[30,33]. The $SO_4^{2-}$ concentrations obtained under light irradiation are much higher than those under dark conditions. This is in consistent with DRIFTS results and further proved the role of DBC in photocatalytic oxidation of $SO_2$ under conditions close to the real atmosphere. To obtain the formation rates of sulfur-containing species on DBC, photooxidation experiments of $SO_2$ (~60 ppb) on DBC of different masses were also carried out (Fig. S5). The measured formation rates of sulfur-containing species varied linearly with the mass concentration of DBC under light irradiation, and the mass normalization rate was determined to be ~$1.0 \times 10^{-3}$ µg h$^{-1}$ (Fig. 1f). On this basis, the formation rates of sulfur-containing products in the wintertime in Beijing due to the photooxidation of $SO_2$ on BC could be estimated to be in the range of 0.01–0.018 µg m$^{-3}$ h$^{-1}$ according to the observed BC mass concentrations (10.4–17.8 µg m$^{-3}$)[32]. This is comparable with the reported formation rate of gaseous $H_2SO_4$ (~0.001–0.1 µg m$^{-3}$ h$^{-1}$) from the OH reaction pathway[27]. These results suggest that the photooxidation of $SO_2$ on BC could be an important source of sulfate in areas with high BC loading.

To further reveal the form of sulfur species on the DBC surface, several characterization techniques were employed. Given that trace amounts of metal elements that can act as catalysts may break off from the $NO_X$ selective catalytic reduction catalyst in diesel after-treatment systems and be emitted along with DBC, we investigated the elemental composition of DBC by XPS. From the survey XPS spectrum of DBC shown in Fig. S6, metallic elements were hardly detected in the DBC. This also implied that the sulfur-containing species detected were not metal sulfates, due to the lack of positive ions containing such metal elements. Thus, we speculated that the photooxidation of $SO_2$ on DBC mainly produces $H_2SO_4$ or sulfur-containing organics rather than metal sulfate complexes.

The pH measurement results in Fig. 2a show that the pH of $SO_2$–aged DBC (~2.68) was apparently lower than that of fresh DBC (~3.74), which suggested that $H_2SO_4$ was formed in the photooxidation reaction on DBC. To confirm the formation of particulate $H_2SO_4$, TGA–MS was employed to characterize $H_2SO_4$ on aged-DBC[40–42]. From the TGA–MS of aged-DBC, a saddle-like change in $SO_2$ (m/z 64 in Fig. 2b) and SO (m/z 48 in Fig. S7) fragments were observed in the temperature intervals of 200–400 and 400–600 °C, respectively. According to the evolution of water and sulfur-containing fragments in the TGA–MS of pure $H_2SO_4$ (Fig. S8), it could be found that the evaporation and pyrolysis of $H_2SO_4$ occur in the temperature range of 200–400 °C. Thus, the formation of $SO_2$ fragments between 200 and 400 °C in the TGA–MS of $SO_2$-aged DBC proved that $H_2SO_4$ exists in $SO_2$–aged-DBC. The $SO_2$ fragments in the range 400–600 °C should be derived from other sulfur-containing species such as $H_2SO_4$–graphite intercalation compounds ($H_2SO_4$-GIC, $(C_{24}^+(HSO_4^-)(H_2SO_4)_2)_n$). Previous studies proved that the pyrolysis of $H_2SO_4$–GIC could produce $SO_2$ fragments in the range 400–600 °C[43,44]. Furthermore, the similar formation and evolution of sulfur-containing fragments in TGA–MS of a $H_2SO_4$–DBC mixture (30 µL $H_2SO_4$ and 15 mg DBC) further support

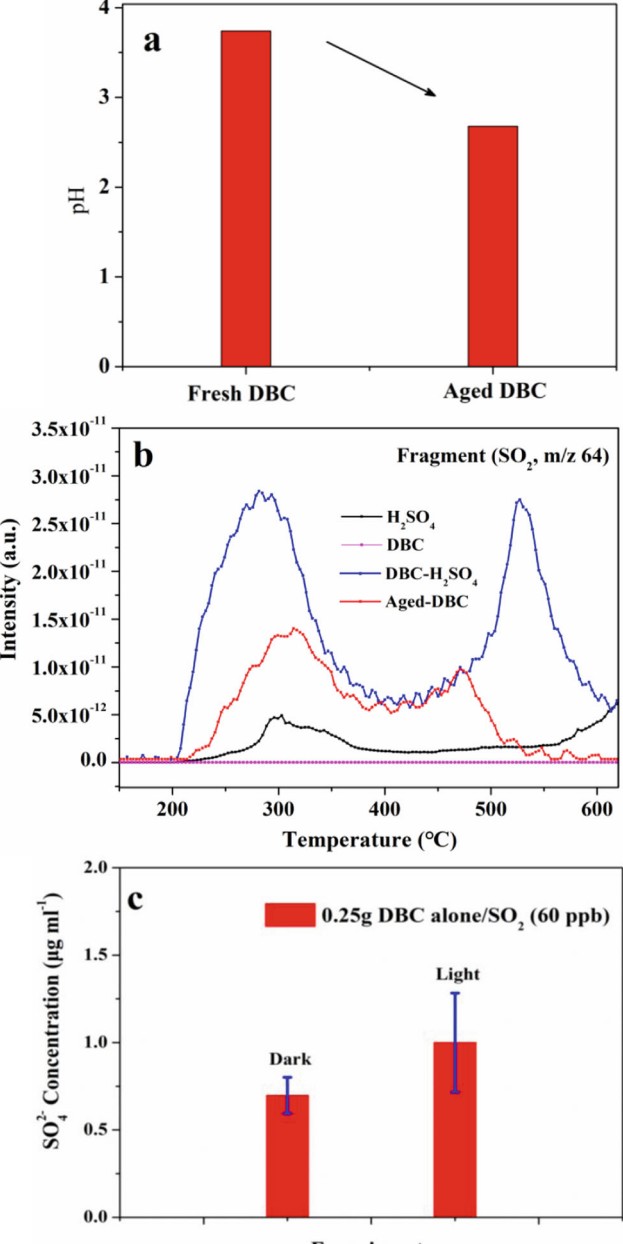

**Fig. 2 | Characterization of $H_2SO_4$ on DBC surface.** pH values of fresh DBC and $SO_2$-aged DBC (**a**). Evaluation and comparison of sulfur-containing fragments (**b**). The comparison of captured gaseous $H_2SO_4$ under dark and light irradiation. Error bars represent standard deviation (**c**).

the supposition that the interaction between DBC and $H_2SO_4$ can produce some other sulfur-containing species (Fig. 2b).

Recent work reported that the soot surfaces upon irradiation can give rise to gaseous OH radicals[45]. It can be speculated that these photoinduced OH radicals may react with $SO_2$ to form gaseous $H_2SO_4$. Thus, to confirm the formation of gaseous $H_2SO_4$, the gaseous products of the photochemical reaction of $SO_2$ on DBC in a quartz photoreaction flow tank (Fig. S9) were captured using formaldehyde solution (4 ml, 11% v/v). As shown in Fig. 2c, the concentration of $SO_4^{2-}$ ions (1.0 ± 0.28 µg ml$^{-1}$) after 24 h reaction under light irradiation was greater than that (0.7 ± 0.1 µg ml$^{-1}$) under dark conditions. This further provided convincing evidence for the formation of gaseous $H_2SO_4$. The condensation of gaseous $H_2SO_4$ could be a source of surface $H_2SO_4$.

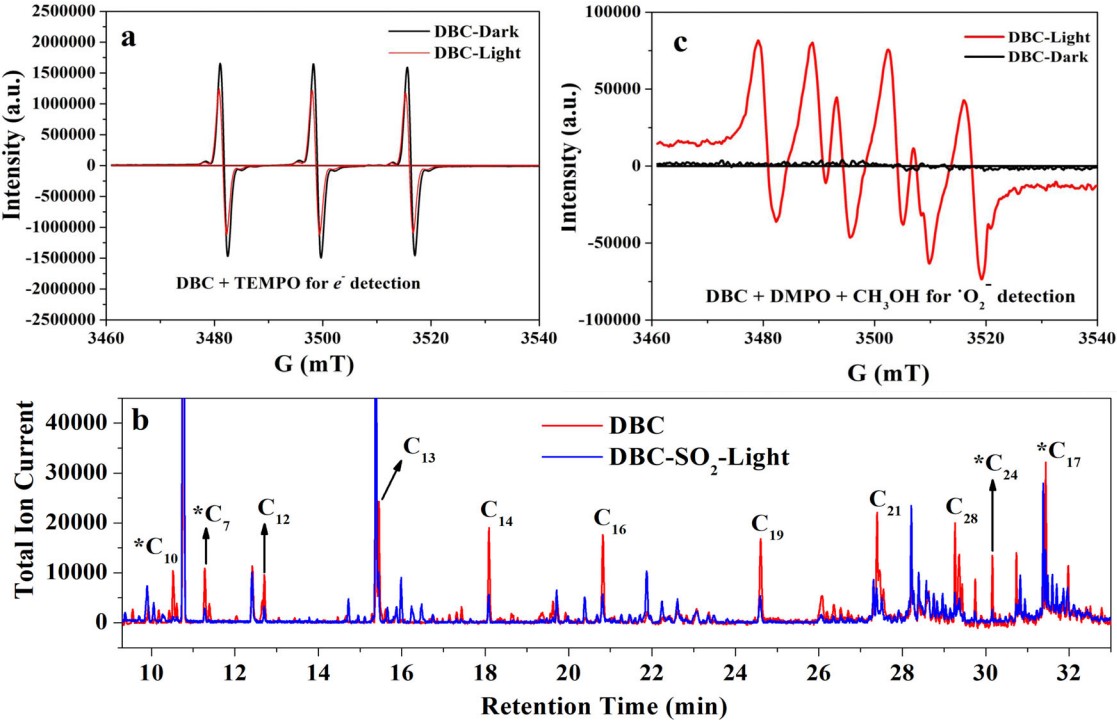

**Fig. 3 | Reactive intermediates on DBC under light irradiation.** TEMPO spin-trapping ESR spectra for the detection of electrons in DBC suspension (**a**). Comparison of OC signals from fresh DBC and DBC aged by $SO_2$ under light irradiation, *C represents aromatic hydrocarbon with different functional groups (such as methyl, oxygen, and chlorine) (**b**). DMPO spin-trapping ESR spectra for the detection of $•O_2^-$ in DBC suspension with methanol ($CH_3OH$) under dark and light irradiation conditions (**c**).

## Heterogeneous formation pathway of $H_2SO_4$

A recent work by Li et al. proved that the EC-initiated photooxidation of soot mainly involves electron transfer and the formation of reactive oxygen species[34]. To shed light on the mechanism of DBC photocatalysis in $SO_2$ oxidation, we also examined the generation of photo-induced electrons and reactive oxygen radicals during DBC photooxidation by the spin-trapping EPR technique. TEMPO was used as the spin-trapping agent for photo-induced electrons. As shown in Fig. 3a, the ESR spectrum of a DBC suspension with TEMPO showed a characteristic signal of three peaks with an intensity pattern of 1:1:1, representing the radical spin-label of TEMPO. It is worth noting that the signal intensity of the TEMPO radical decreased once exposed to light irradiation, demonstrating that TEMPO was partially reduced to an ESR-silent molecule such as TEMPOH by photo-induced electrons[46,47]. The conversion of TEMPO to TEMPOH indirectly indicated that DBC photooxidation can indeed induce the generation and transfer of electrons.

Li et al. reported that OC in soot is the major donor of electrons during soot photoaging[34]. To verify this, the OC in DBC was extracted and further analyzed by GC-MS. The results of GC-MS (Fig. 3b) showed that OC extracts from DBC mainly consist of saturated long-chain saturated alkanes with more than 10 carbons, and aromatic hydrocarbons with different functional groups (such as methyl, oxygen, and chlorine). It is worth noting that the relative signal intensity of OC significantly decreased after DBC was photoaged under light irradiation. This indicated that light irradiation should result in the photooxidation of OC in DBC and also implied that OC maybe the crucial donor of electrons during DBC photooxidation. However, it should be noted that there was almost no absorbance in the range of 200–800 nm for extracted OC as determined by diffuse reflectance UV–vis spectroscopy (Fig. S10b). This indicated that the long-chain saturated alkanes could not directly donate electrons under light irradiation. In contrast to OC, EC in DBC distinctly adsorbed radiation over a broad wavelength range from 200 to 800 nm (Fig. S10b).

Several studies reported that carbonaceous materials excited under light irradiation can induce the formation of surface electron–hole pairs, especially for these surfaces with plentiful surface structure defects and oxygen functional groups[37,48,49]. Given that various oxygen functional groups and structural defects or disordered structures are ubiquitous on carbonaceous materials in DBC (Figs. S1a and S11), the formation of photo-generated holes ($h^+$) on excited EC may abstract electrons from OC and subsequently donate electrons to other available acceptors such as adsorbed $O_2$ (Eq. (3))[50–53]. To further verify this, the photo-induced electron–hole pairs in residual EC were analyzed using TEMPO spin-trapping ESR spectra. As shown in Fig. S12, the remarkable decrease in the signal intensity of TEMPO radicals after 120 min light irradiation further demonstrates that the residual EC from DBC can indeed induce the generation of holes or electrons (Eq. (1)). Thus, long-chain saturated alkanes in DBC can indirectly donate electrons to absorbed $O_2$ via excited EC (Eqs. (2) and (3)) and subsequently be oxidized to other organic oxygen-bearing compounds (Eq. (4))[34].

To verify whether the observed photogenerated electrons can further result in the formation of reactive oxygen radicals ($•O_2^-$ and OH radical), we chose two spin traps, DMPO and BMPO, to capture $•O_2^-$ and OH radical, respectively. As shown in Fig. 3c, four characteristic peaks of DMPO–$•O_2^-$ adducts were observed in a DBC suspension in methanol under light irradiation, while no signal was observed under dark conditions[54]. This demonstrated that electrons were transferred to the dissolved $O_2$ in the DBC suspension, resulting in the formation of $•O_2^-$ radical (Eq. (3)). Previous studies reported that $•O_2^-$ radicals can interact with $SO_2$ or sulfite ion ($SO_3^{2-}$) to form a series of sulfur-containing radicals ($SO_3^{•-}$, $SO_5^{•-}$, and $SO_4^{•-}$)[55–57]. The free radical chain reaction dominated by sulfur-containing radicals would eventually result in sulfate formation. However, it should be noted that these sulfur-containing radicals are scarcely detected in the BMPO spin-trapping ESR spectra of $SO_2$-aged DBC (Fig. 4a) according to the reported measurement methods[58,59]. Thus, the contribution of the

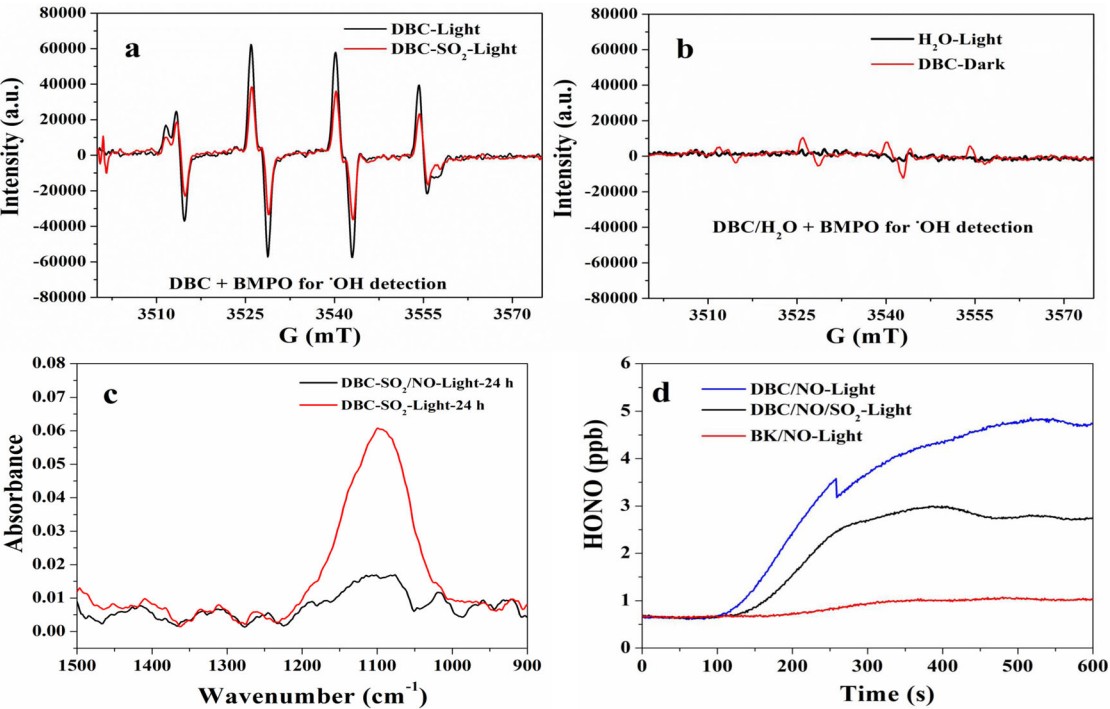

**Fig. 4 | Photo-induced OH radical promoting the heterogeneous conversion of SO₂ to H₂SO₄.** BMPO spin-trapping ESR spectra for the detection of OH radical in DBC suspension under light irradiation (**a**). BMPO spin-trapping ESR spectra for the detection of OH radical in DBC suspension in the dark and in water under light irradiation (**b**). Comparison of in-situ DRIFTS spectrum of sulfate in the absence and presence of 10 ppm NO (**c**). The heterogeneous conversion of NO (168 ppb) to HONO under different conditions (**d**).

heterogeneous reaction between SO₂ and •O₂⁻ to H₂SO₄ should be limited despite the fact that this reaction may occur in this system.

In addition to its direct oxidation capability, •O₂⁻ radical is also the key intermediate species in OH radical production during photo-induced interfacial reactions; thus, OH radicals in DBC suspension were further measured via the spin trapping EPR technique[46,60]. As shown in Fig. 4a, a significant signal of BMPO−OH adducts with a typical 1:2:2:1 quartet signal was observed, indicating that light irradiation could indeed cause the formation of OH radicals in the suspension of fresh DBC and DBC aged by SO₂. Almost no BMPO−OH adduct signals were observed in DBC under dark conditions or in ultrapure water under light (Fig. 4b). This further indicated that OH radical formation was closely related to the presence of DBC and light irradiation. These results proved that the photooxidation process of DBC indeed involves a complex electron transfer pathway and results in the generation of reactive oxygen radicals such as OH radicals (Eqs. (3), (5), and (6)). Moreover, the BMPO−OH adduct detected in aged-DBC treated with SO₂ apparently decreased relative to that in fresh DBC (Fig. 4a). This implied that the heterogeneous conversion of SO₂ to H₂SO₄ (Eqs. (7) and (8)) should be related to the consumption of OH radical from DBC. To further confirm this, the heterogeneous conversion of SO₂ on DBC under light irradiation was also investigated in the presence of high-level NO (radical scavenger). From the results of in-situ DRIFTS experiments shown in Fig. 3c, the presence of high-level NO (10 ppm) could significantly suppress the heterogeneous formation of H₂SO₄ on DBC. Moreover, the results of experiments in a coated-wall quartz flow tube reactor also showed that the presence of DBC indeed can promote the conversion of NO to HONO under light irradiation relative to the control experiment. Furthermore, the addition of SO₂ can also in turn hamper the heterogeneous conversion of NO to HONO (Fig. 4d). These results supported the hypothesis that the competing reactions of SO₂ + OH and NO + OH must exist during DBC photooxidation. This also proves that the heterogeneous oxidation of SO₂ by OH radicals should be an important formation pathway of

particulate H₂SO₄ on DBC under light irradiation.

$$EC \xrightarrow[\text{excited}]{h\nu} h^+ + e^- \tag{1}$$

$$OC + h^+ \rightarrow OC^+ \tag{2}$$

$$O_2 + e^- \rightarrow {}^\bullet O_2^- + H^+ \rightarrow HO_2^\bullet \tag{3}$$

$$OC^+ + {}^\bullet O_2^- / {}^\bullet OH \rightarrow \text{Carbonyl or others} \tag{4}$$

$$2HO_2^\bullet \rightarrow O_2 + H_2O_2 \tag{5}$$

$$H_2O_2 \xrightarrow{h\nu} 2{}^\bullet OH \tag{6}$$

$${}^\bullet OH + SO_2 + O_2 \rightarrow SO_3 + HO_2^\bullet \tag{7}$$

$$SO_3 + H_2O \rightarrow H_2SO_4 \tag{8}$$

## Discussion

The conventional view recognizes BC particles as a reducing agent in the atmosphere. For example, BC was found to initiate the heterogeneous reduction of NO₂ to HONO and hence elevate the atmospheric oxidation capacity in the atmosphere[19,20,22]. In this work, our experimental results indicated that DBC under light irradiation can act as an oxidation medium to directly promote the heterogeneous oxidation of SO₂ to H₂SO₄. Moreover, it was also proven that the promoting role of DBC photochemistry on the rapid conversion of SO₂ is linked to the photo-induced formation of OH radicals. As for the

source of OH radicals on soot under light irradiation, He et al. proposed that these OH radicals mainly derive from the reaction of photoinduced singlet oxygen ($^1O$) and $H_2O$ on soot based on the density functional theory (DFT) results[45]. In this study, we found that the formation of OH radical is also closely related to the formation and conversion of the intermediate reactive superoxide radical ($\cdot O_2^-$) triggered by the abstraction and transfer of photoinduced electrons. Thus, our work further complements or improves the production mechanism of OH radicals on DBC under light irradiation, which provides insight into the photochemical reaction process on DBC.

$H_2SO_4$ in the atmosphere mainly comes from the reaction of $SO_3$ and $H_2O$, which is the critical precursor causing the rapid nucleation and growth of ultra-fine particle[61,62]. Thus, exploring the unknown sources of $SO_3$ and $H_2SO_4$ is crucial for understanding the formation and growth of new particles. Recently, both DFT calculation and field observation indicated that the formation of gaseous $SO_3$ during the early morning is closely related to the catalytic oxidation of $SO_2$ on the surface of soot[30,31,33]. Another field observation by Zhang et al. found that the photooxidation of BC-containing particles can further enhance the formation of sulfate[38]. Here, our experimental results provide reliable experimental evidence for these recent observations and DFT calculation results, especially under light irradiation. Hence, the heterogeneous oxidation of $SO_2$ to $H_2SO_4$ driven by the photochemical process on DBC may very directly contribute to the rapid formation and growth of new particles in the atmosphere via inducing the formation of gaseous and particulate $H_2SO_4$. Our study highlights the photooxidation role of DBC in the heterogeneous formation of $H_2SO_4$, which has important atmospheric implications for understanding new particle formation and the source of atmospheric sulfate. It was worth noting that further model simulation and field observation in future studies should be effectively combined to quantitatively evaluate the contribution of this new pathway to $H_2SO_4$ formation in the atmosphere.

Additionally, black carbon has strong effects on regional and global climate due to the remarkable positive (warming) radiative forcing in the atmosphere[63–65]. In particular, the internal mixing between BC and other aerosol components through processes such as gas condensation and coagulation can remarkably affect light absorption by BC[12,66,67]. This work proved that the photooxidation of DBC could directly promote the heterogeneous conversion of $SO_2$ to $H_2SO_4$. Thus, an in-depth study of the optical properties of BC aerosol internally mixed with sulfuric acid in the future will help to evaluate the effect of variation of the mixing state on direct radiative forcing and climate.

## Methods
### Experimental procedure
Diesel soot was collected from the diesel particle filter (DPF) of a China VI heavy-duty diesel engine (ISUZU from China). A diesel engine bench test was run under the conditions of the World Harmonized Transient Cycle (WHTC). China VI fuels were used in this study, meeting the GB T32859-2016 standard. The heterogeneous reactions of $SO_2$ or $SO_2/NO$ on diesel soot particles were measured by in-situ DRIFTS (NEXUS 6700, Thermo Nicolet Instrument Corporation), equipped with a diffuse reflection chamber and a high-sensitivity mercury cadmium telluride (MCT) detector. The MCT was cooled by liquid $N_2$ prior to the measurement. The infrared spectra were collected by means of a computer using OMNIC 6.0 software (Nicolet Corporation, USA). All spectra reported here were recorded at a resolution of $4\,cm^{-1}$ for 100 scans in the spectral range $4000–650\,cm^{-1}$. The spectra are presented in the Kubelka−Munk (K−M) scale, which can provide a better linear relation with concentration via reducing or eliminating the mirror effect. To simulate solar irradiation, a high uniformity integrated xenon lamp (PLS-FX300HU, Beijing Perfectlight Technology Co., Ltd.) was used as the light source[34]. Its visible spectrum ranges from 330 to 850 nm (Fig. S10a). The light in the near-infrared and infrared bands was filtered

using a transmission-reflection filter (VISREF). DBC was placed into a cylindrical ceramic crucible in the diffuse reflection chamber before DRIFTS measurement. Before the reaction, the DBC was purged with $200\,mL\,min^{-1}$ air at 298 K and 55% RH until the infrared spectrum was unchanged. Then the samples were exposed to 10 ppm $SO_2$ balanced with $200\,mL\,min^{-1}$ synthetic air for at least 12 h.

The experiments on HONO detection were conducted in a coated-wall quartz flow tube reactor (34 cm length, 1.6 cm i.d.)[68,69]. The coated tube with the deposited DBC sample was horizontally placed in the main reactor. Synthetic air, as the carrier gas, introduced NO and $SO_2$ into the coated flow tube at a total flow rate of $2\,L\,min^{-1}$. Inorganic gases ($SO_2$ and NO) were introduced into the flow tube through a movable injector with 0.3 cm radius. HONO was measured by a long-path absorption photometer (HONO-1000, Beijing Zhichen Technology Co. Ltd), while the reactant gas $SO_2$ in the coated flow tube was measured by a Thermo Scientific analyzer (43i $SO_2$ analyzer). To better simulate the heterogeneous conversion of $SO_2$ on DBC under close-to-atmosphere conditions, heterogeneous experimental tests with low $SO_2$ concentrations (∼60 and 10 ppb) were further carried out in a quartz tube plug flow reactor (40 cm in length, 0.6 cm in diameter) at 37% RH. Before loading the sample into the plug flow reactor, ∼2 mg DBC needed to be diluted using ∼1 g silica sand (analytically pure, 1–2 mm in diameter) to ensure the reaction gas could flow fluently through the DBC. To further verify that DBC photooxidation in the presence of $SO_2$ could also promote the formation of gaseous $H_2SO_4$, the heterogeneous reaction of $SO_2$ (∼60 ppb) on DBC (∼0.25 g) powder alone was carried out in a quartz photoreaction tank at 50% RH. Prior to the reaction, 0.25 g DBC power was uniformly dispersed into a shallow quartz vessel (7 cm in diameter and 0.4 cm in depth) and then placed into the photoreaction tank. The outlet of the quartz photoreaction tank was connected with a quartz trap bottle (2 cm in diameter and 7 cm in depth) with 4 ml formaldehyde solution (20% v/v). Gaseous $H_2SO_4$ from the interaction between $SO_2$ and DBC was captured using ultrapure water in the quartz trap. The captured $H_2SO_4$ was analyzed using IC. The designed RH in the DRIFTS chamber or in the flow tube was obtained by varying the ratio of dry zero air to wet zero air. A flow of humid vapor was generated by bubbling zero air through ultrapure water. A Vaisala HMP110 probe was used to monitor the changes in RH online.

### DBC characterization
Surface chemical states were analyzed by X-ray photoelectron spectrometry (XPS) (ESCALAB 250Xi, Thermo Scientific). An X-ray powder diffractometer (Bruker D8 ADVANCE 2θ diffractometer) with Cu Kα radiation ($\lambda$ = 0.15406 nm) operated at 40 kV and 40 Ma was used to characterize the crystalline form and interlayer spacings of the DBC power. The patterns were measured over the 2θ range from 10° to 80° with a scanning step size of 0.02°. The OC in DBC was analyzed and identified via gas chromatography coupled with mass spectrometry (GC−MS, Agilent 6890−5973). DBC was first ultrasonically extracted for 10 min using 20 ml of dichloromethane ($CH_2Cl_2$), which was filtered through a quartz sand filter. The obtained supernatant liquid was subsequently concentrated using the $N_2$ blowing method for final analysis. The gas chromatograph was equipped with a DB-5MS $30\,m \times 0.25\,mm \times 0.25\,mm$ capillary column and the mass spectrometer employed a quadrupole mass filter with a 70 eV electron impact ionizer. The temperature of the programmed temperature vaporizer was held at 270 °C. The initial oven temperature was set at 40 °C for 2 min, then increased step-by-step to 150 °C (by 5 °C $min^{-1}$) for 5 min, 280 °C (by 10 °C $min^{-1}$) for 10 min, and 320 °C (by 10 °C $min^{-1}$) for 5 min. A pH meter (Mettler pH, S220-K) was used to measure the acidity change of DBC after the reaction. The UV−vis spectra of residual EC and extracted OC were measured using the UV−vis spectroscopy (Perkin Elmer LAMBDA 650).

**Analysis and characterization of oxidizing agents and products**

**Thermal gravimetric analysis.** The sulfur-containing species produced during diesel soot photooxidation were investigated using a combined method of thermogravimetry–mass spectrometry (TGA–MS). In brief, a Mettler-Toledo thermogravimetry system (TGA, DSC1-1600HT) was coupled with a quadrupole mass spectrometer (MS, ThermoStar- GSD/350, Pfeiffer Vacuum) by a silica capillary at a temperature of 250 °C. The MS system was equipped with an electron ionization source with the voltage at 70 eV and provided the mass spectra up to $m/z$ 300. TGA–MS was carried out over the range 35–800 °C with a 30 K min$^{-1}$ gradient. The whole experiment was accomplished under an inert purge gas of $N_2$ with a constant flow rate of 70 ml min$^{-1}$. In a typical desorption run, a blank test of the empty sample crucible was performed at 25 °C in an $N_2$ stream (the carrier gas flow rate: 50 mL min$^{-1}$ and the shielding gas flow rate: 20 mL min$^{-1}$). Afterward, the sample was weighed and placed in the sample container. The pure DBC (15 mg), $SO_2$-aged DBC (15 mg), pure $H_2SO_4$ solution (60 μL), and an $H_2SO_4$–DBC mixture (30 μL $H_2SO_4$ (1.8 mol/L) and 15 mg DBC) were placed in sequence in an alumina crucible of 70 μL. The mass spectra of $m/z$ 98, 80, 64, and 48, which correspond to the main fragment components of $H_2SO_4$ ($H_2SO_4$, $SO_3$, $SO_2$, $SO$, respectively), were chosen to further verify $H_2SO_4$ formation.

**ESR measurements.** An electron spin resonance spectrometer (ELEXSYS E500 ESR; Bruker) with a modulation frequency of 100 kHz and a microwave frequency of 9.5 GHz was used to capture reactive radical signals. 5-tert-butoxycarbonyl-5-methyl-1-pyrroline-N-oxide (BMPO) added to a freshly prepared 0.1 M solution in deionized water was used for capturing OH radicals in aqueous solutions. 5,5-dimethylpyrroline-N-oxide (DMPO) was used to capture superoxide ($\cdot O_2^-$). 2,2,6,6–tetramethylpiperidine-1-oxyl (TEMPO) was used to capture the photo-induced electrons in the DBC suspension. DBC suspensions loaded in a quartz capillary tube (1 mm Ø and 10 cm length) were continuously irradiated with the Xenon lamp during the monitoring of radical signals in the ESR spectrometer. The typical parameters for ESR measurement were as follows: the sweep width was 100 G, the modulation amplitude was 1.00 G, and the sweep time was 81.92 ms.

**IC analysis.** To quantitively the $H_2SO_4$ concentration under different conditions, Ion chromatography (IC, Model DIONEX ICS-2100, Thermo Scientific, Inc., USA) was employed to analyze the changes in the concentrations of water-soluble ions (such as $SO_4^{2-}$ and $NO_3^-$). A certain amount of $SO_4^{2-}$ may be contained in the primordial DBC samples due to the combustion of sulfur-containing species in fuel. Thus, DBC samples were ultrasonically extracted using ultrapure water at least 10 times to exclude interference from these preexisting $SO_4^{2-}$ ions in the measurement of newly produced $SO_4^{2-}$ ions during IC analysis. After the reaction finished, DBC samples (25 mg) obtained under different experimental conditions were first dispersed in 20 mL of ultrapure water and sonicated for 10 min, and subsequently filtered using a syringe filter (13 mm diameter, 0.22 μm pore-size). The filtered liquid was split in half. Half of the samples were directly analyzed by IC.

## Data availability

The data that support the findings of this study are available from the corresponding author upon reasonable request. Source data are provided with this paper.

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

## Acknowledgements

This work was financially supported by the National Natural Science Foundation of China (grant nos. 22188102, 21976098, 21922610, 41877304, and 22006152). Thanks are due to Yan Zhao [Nanjing University of Information Science & Technology, China] for her instructive suggestions. We also thank Qingcai Feng for her great help in editing and polishing this paper.

## Author contributions

H.H. designed and supervised the research. P.Z. designed and performed the experiments. Q.M. gave guidance for the experimental research. T.C., Q.M., B.C., Y.W., Y.M, and Y.Y. provided suggestions for the manuscript. P.Z. and H.H. wrote the manuscript. All authors discussed the results and commented on the manuscript.

## Competing interests

The authors declare no competing interests.
