## [Peer Review File · Nature Communications]

Title: Diesel soot photooxidation enhances the heterogeneous formation of H₂SO₄REVIEWER COMMENTS

Reviewer #1 (Remarks to the Author):

The topic is interesting and very important.

However, before acceptance it is important to answer following items.

Major:

The MS is qualitative, and qualitatively sound. However, it would be good to show some quantitative numbers e.g. how much sulphuric acid can be produced e.g. in atmospheric conditions with different BC concentrations. This can then be compared to gas phase production of sulphuric acid.

Minor:

line 123, it is said that SO₂ concentration is in low level (1 ppm). It is very high concentration, typical atmospheric concentrations are well below 1 ppb.

In the main text there are plenty of acronyms, which are not explained like TEMPO in line 182

Lines 270-272, atmospheric NPF is discussed without references. Several references can be add e.g. Kulmala et al., 2021, Faraday Discussion, Lei et al., 2018 Science.

Reviewer #2 (Remarks to the Author):

This is an interesting new contribution aiming at unravelling the chemistry that leads to SO₂ photoconversion on soot under atmospheric conditions. This is an important topic and is of wide interest, especially in Asian megacities subject to intense haze episodes.

The manuscript stays quite qualitative and some time vague. For instance, what is meant with “internal catalytic mechanism” (Line 34), or “expose” (line 100, where it is probably a mis use of this word), “inorganic sulfate” (line 132, H₂SO₄ is inorganic). It is clearly shown that SO₂ reacts on the surface of soot, even if the actual speciation of the products is only made indirectly, and that light induce the formation of transient oxidants.

The reasoning leading to the conclusion that OH is a key player in the current observations is unclear to this reviewer. O₂⁻ could certainly play a similar role, while OH may be scavenged by the OC fraction on soot, the superoxide may react with both NO and SO₂ (and not so much with the alkanes and aromatics). Any thoughts on this?

It is stated that the irradiation takes place above 350 nm, but there is no mention how the Xenon is filtered to achieve such wavelength. Please add this info. This is indeed a critical point. In fact, such wavelength region limits substantially the nature of compounds that may absorb light and trigger the discussed photochemistry. It certainly means that the long chain saturated alkanes are not involved in the discussed chemistry. It also means that most of the simple aromatics are not triggering the observations. Which contradicts several statements made in this manuscript concerning the suggested mechanism. Maybe the author should elaborate a bit more on the actual compounds that may indeed

act as an electron source (i.e., reaction 1).

Reviewer #3 (Remarks to the Author):

Ms.No. NCOMMS-22-11710

Zhang et al. are presenting results of an experimental study on heterogeneous SO₂ oxidation carried out on irradiated soot / black carbon that originates from a Diesel engine (DBC). Analysis was conducted by means of DRIFTS following the IR feature, which is attributed to sulfate formation. The authors try to follow from the observed sulfate signal that sulfuric acid is formed. Reactive intermediates were measured by spin-trapping and a reaction scheme of heterogeneous SO₂ oxidation on DBC is proposed. This manuscript is an interesting and timely work and is well written. New pathways of SO₂ oxidation leading to sulfuric acid (SA) represent a hot topic in atmospheric sciences. Possible heterogeneous processes in this context are under debate since a couple of years.

Here my comments:

- 1) In the Abstract etc. it sounds like the authors mean that DBC-based SO₂ oxidation could lead to additional gas-phase SA in the atmosphere, right? Or do they mean only surface SA? Please clarify. No experimental proof of gas-phase SA formation is presented for atmospheric conditions! Moreover, DRIFTS only shows the formation of sulfate at the surface. The argumentation that it is likely SA at the surface (in part) is not convincing to me.
- 2) The desorption/pyrolysis results in Fig. 1, panel D, are based on total traces of SO and SO₂ fragments, which are not specific for SA. At temperature higher than 350°C, SA starts to decompose forming SO₃ and H₂O. How does it influence data analysis? The TGA-MS experiments are not described in Methods.
- 3) Yao et al., *ES&T Lett.* (2020), 10.1021/acs.estlett.0c00615, proposed SO₃ formation from soot-based SO₂ oxidation supported by direct SO₃ measurements in the atmosphere. Can the authors rule out a similar process here? Please discuss this topic.
- 4) Line 123: Atmospheric SO₂ levels are a few ppbv. In the experiments, “low SO₂” was about 1 ppm, i.e. about a factor of 1000 higher. Is the heterogeneous SO₂ conversion that slow that such high SO₂ is needed? What does it mean for the reaction rate for atmospheric conditions and its importance for global SO₂ oxidation?
- 5) Line 177, spin-trapping: It is nice to have these results. O₂- and OH are the reactive intermediates, well in line with that what can be expected from knowledge in literature. Or can we draw any special conclusion for this system here?

All in all, it is a nice piece of work in the field of heterogeneous SO₂ oxidation. But I see not special novelty value or it is not clearly communicated. If the authors mean that the described process is important for atmospheric SA production, gas-phase SA measurements for close to atmospheric conditions should be added to verify this hypothesis.

The manuscript in its present form is more suited for a special journal dealing with heterogeneous catalysis in the atmosphere or atmospheric chemistry generally.

Point-by-Point Response to the Reviewers' Comments

Responses to Comments from Reviewer #1:

The topic is interesting and very important. However, before acceptance it is important to answer following items.

Answer: Thank you very much for giving us these valuable comments. The manuscript has been revised thoroughly according to your comments.

Major:

Q1. The MS is qualitative, and qualitatively sound. However, it would be good to show some quantitative numbers e.g. how much sulphuric acid can be produced e.g. in atmospheric conditions with different BC concentrations. This can then be compared to gas phase production of sulphuric acid.

A1. Thank you for your suggestions. To quantitatively evaluate the role of DBC in H₂SO₄ formation rates under conditions close to the real atmosphere, photooxidation experiments of SO₂ (~60 ppb) on DBC with different mass concentrations were carried out in a flow reactor. The schematic diagram of experimental setup is shown in Fig. R1. It was found that SO₂ uptake increases apparently with the DBC mass concentration (Fig. R2A). The measured H₂SO₄ formation rates varied linearly with the mass concentration of DBC under light irradiation (Fig. R2B) and the mass normalization rate was determined to be $\sim 1.0 \times 10^{-3} \text{ h}^{-1}$. BC mass concentrations have been observed to range from 10.4 to 17.8 $\mu\text{g m}^{-3}$ during haze events.¹ On this basis, H₂SO₄ formation rates in the real atmosphere were estimated to be in the range of 0.01-0.018 $\mu\text{g m}^{-3} \text{ h}^{-1}$ according to the obtained linear equation in this work. This is comparable to the formation rate of gaseous H₂SO₄ (~ 0.001 - 0.1 $\mu\text{g m}^{-3} \text{ h}^{-1}$) from the OH reaction pathway under relatively clean conditions.² These results suggest that the photooxidation of SO₂ on the surface of black carbon could be an important source of sulfate in areas with high black carbon loading.

This discussion has been added into the revised manuscript (Lines 148158).

Fig. R1. Schematic diagram of the flow reactor setup.

Fig. R2. SO₂ uptake on DBC of different mass concentrations (A); H₂SO₄ formation rate as a function of DBC concentration (B).

Q2. line 123, it is said that SO₂ concentration is in low level (1 ppm). It is very high concentration; typical atmospheric concentrations are well below 1 ppb.

A2. In China, the average SO₂ concentration was in the range of 50 - 75 ppb from 2012 to 2014.³ In the last few years, the average SO₂ concentration has exhibited a constant decrease due to the reduction in the emission of SO₂. The field observation found that

the hourly average concentration of SO₂ was in the range of 2-12 ppb during the period of winter residential heating in Beijing (Fig. R3). Thus, to simulate the heterogeneous conversion of SO₂ on DBC surface under close to real atmospheric conditions, photooxidation experiments of low-level SO₂ (10 ppb and 60 ppb) on DBC were carried out in the flow reactor. Fig. R4A shows the total uptake of 60 ppb SO₂ on particles under different conditions. When SiO₂ particles in the tube plug flow reactor were exposed to SO₂ (60 ppb), SO₂ concentrations decreased slightly and returned to the initial concentration in 8 min. This indicated that the reaction of SO₂ on the fresh SiO₂ surface is weak. In contrast, a sharp decrease in the SO₂ concentration was detected in the DBC/SiO₂ system under either dark conditions or light irradiation. The uptake process lasted much more than 10 h. Under light irradiation, the total SO₂ uptake on DBC is slightly greater than that under dark conditions. These results indicate that the irreversible uptake of SO₂ on DBC is significant even under conditions close to the real atmosphere. To further quantitatively assess the heterogeneous conversion of SO₂ on DBC, extracted SO₄²⁻ ions were analyzed using IC (Fig. R4B). As can be seen, the SO₄²⁻ concentration is the highest (3.7±0.59 μg ml⁻¹ for 60 ppb SO₂) in the presence of DBC under light irradiation compared with the control experiments. This further proved that both light irradiation and DBC have a significant enhancing role on the heterogeneous conversion of SO₂. Moreover, the higher SO₄²⁻ concentration obtained at 10 ppb SO₂ (1.5±0.76 μg ml⁻¹ for 10 ppb SO₂) further highlights the enhancing role of DBC photooxidation at lower SO₂ levels.

Fig. R3. The hourly average concentration of SO₂ from 30 July 2020 to 20 May 2021 in Beijing.

Fig.R4. SO₂ uptake on DBC under different conditions (A); comparison of H₂SO₄ concentrations under different conditions (B).

This discussion has been added into the revised manuscript (Lines 133-147).

Q3. In the main text there are plenty of acronyms, which are not explained like TEMPO in line 182

A3. The acronyms in the main text have been explained in the revised manuscript.

For example, TEMPO represents the acronym for 2,2,6,6-tetramethylpiperidine-1-oxyl; DMPO is the acronym for 5,5-Dimethylpyrroline-N-oxide; BMPO is the acronym for 5-tert-butoxycarbonyl-5-methyl-1-pyrroline-N-oxide (BMPO). DRIFTS is the acronym for *in-situ* diffuse reflectance infrared Fourier transform spectroscopy.

The corresponding full names for these acronyms have been added in the “Methods” section (Lines 428-431).

Q4. Lines 270-272, atmospheric NPF is discussed without references. Several references can be added e.g. Kulmala et al., 2021, *Faraday Discussion*, Lei et al., 2018 *Science*.

A4. Thank you for your reminder. The corresponding references from Kulmala et al., (*Faraday Discussion*, 2021, **226**, 334-347) and Lei et al., (*Science*, 361.6399 (2018): 278-281) have been cited in the discussion of atmospheric NPF in the revised manuscript (Line 317).

Responses to Comments from Reviewer #2:

This is an interesting new contribution aiming at unravelling the chemistry that leads to SO₂ photoconversion on soot under atmospheric conditions. This is an important topic and is of wide interest, especially in Asian megacities subject to intense haze episodes.

Answer: Thank you very much for giving us these valuable comments and suggestions. The manuscript has been carefully revised according to your comments (See below). The manuscript stays quite qualitative and some time vague.

Q1. For instance, what is meant with “internal catalytic mechanism” (Line 34), or “expose” (line 100, where it is probably a mis use of this word), “inorganic sulfate” (line 132, H₂SO₄ is inorganic). It is clearly shown that SO₂ reacts on the surface of soot, even if the actual speciation of the products is only made indirectly, and that light induce the formation of transient oxidants.

A1. Thank you very much for pointing out the mistakes in our writing. These mistakes or typos are corrected as follows.

(1) “internal catalytic mechanism” has been revised to “catalytic mechanism” (Line 34).

(2) “expose” has been revised to “explore” (Line 100).

(3) “inorganic sulfate” has been corrected to “metal sulfate” (Line 165).

These errors have been corrected in the revised manuscript.

Q2. The reasoning leading to the conclusion that OH is a key player in the current observations is unclear to this reviewer. O_2^- could certainly play a similar role, while OH may be scavenged by the OC fraction on soot, the superoxide may react with both NO and SO_2 (and not so much with the alkanes and aromatics). Any thoughts on this?

A2. We agree with you that O_2^- may play a role in these reactions on DBC. Previous studies showed that $\bullet O_2^-$ can interact with SO_2 or sulfite ion (SO_3^{2-}) to form a series of sulfur-containing radicals ($\bullet SO_3^-$, $\bullet SO_5^-$, and $\bullet SO_4^-$).⁴⁻⁶ These radical chain reactions dominated by sulfur-containing radicals would eventually result in the formation of sulfate. However, it should be noted that these sulfur-containing radicals are scarcely detected in the BMPO spin-trapping ESR spectra of SO_2 -aged DBC (Fig.3A) according to the reported measurement methods.^{7, 8} Thus, we concluded that the contribution of heterogeneous reaction between SO_2 and O_2^- to form H_2SO_4 /sulfate should be limited despite the fact that this reaction may happen in this reaction system.

This discussion has been added into the revised manuscript (Lines 248-256).

Q3. It is stated that the irradiation takes place above 350 nm, but there is no mention how the Xenon is filtered to achieve such wavelength. Please add this info. This is indeed a critical point. In fact, such wavelength region limits substantially the nature of compounds that may absorb light and trigger the discussed photochemistry. It certainly means that the long chain saturated alkanes are not involved in the discussed chemistry. It also means that most of the simple aromatics are not triggering the observations. Which contradicts several statements made in this manuscript concerning the suggested mechanism. Maybe the author should elaborate a bit more on the actual compounds that may indeed act as an electron source (i.e., reaction 1).

A3. (1) The Xenon spectrum is shown in Fig. R5A. The wavelength region of the Xenon spectrum is in the range of 330 to 850 nm. Without filtering, light with wavelengths less than 330 nm was not detected. The light in the near-infrared and infrared bands was filtered using a transmission-reflection filter (VISREF). This information was added in the revised manuscript (Lines 354-358).

Fig. R5. Visible spectrum of Xenon light (A); UV-vis spectra of residual EC and extracted OC (B).

(2) We agree with you that organic compounds could not absorb light in this region. In fact, the absorption characteristics of OC (extracted with 15 ml CH_2Cl_2 from ~0.5 g DBC) and EC in DBC measured with UV-vis spectroscopy (PerkinElmer LAMBDA 650) showed that there was almost no absorbance in the range of 200 to 800 nm for the extracted OC, whereas EC absorbed radiation over two broad wavelength ranges, from 200 to 300 nm and from 380 to 800 (Fig. R5B). This confirms that the long-chain saturated alkanes alone could not directly donate electrons via the absorption of light. However, OC in DBC may participate in reactions through other pathways.

Several studies reported that carbonaceous materials excited under light irradiation can induce the formation of surface electron-hole pairs, especially for these surfaces with plentiful defects and oxygen-containing functional groups.⁹⁻¹¹ Given that various oxygen-containing functional groups (Fig. R6) and defects or disordered structures (D peak in Fig.R7) are ubiquitous on carbonaceous materials in DBC, it seems reasonable that the formation of photo-generated holes on excited EC may extract electrons from OC and subsequently donate electrons to other available acceptors such as adsorbed O₂.¹²⁻¹⁵ To further verify this assumption, the photo-induced electron-hole pairs induced by the illumination of EC were analyzed using TEMPO spin-trapping ESR spectra. As shown in Fig.R8, the remarkable decrease in the signal intensity of the TEMPO radical after 120 min light irradiation demonstrated that EC in DBC samples can indeed induce the generation of holes or electrons (Eq-1). Thus, long-chain saturated alkanes in DBC could react with photo-induced holes on excited EC (Eq-2) and subsequently be oxidized to other organic oxygen-bearing compounds (Eq-4).¹⁶ The electrons contributing to the formation of •O₂⁻ were the photo-induced electrons (Eq-3), which could also be considered to be indirectly derived from OC.

The reaction mechanism has been corrected as follows:

Fig. R6. *in-situ* DRIFT spectra of DBC.

Fig. R7. Raman spectrum of DBC sample.

Fig. R8. TEMPO spin-trapping ESR spectrum for the detection of electrons in residual EC suspension

These discussions have been added in the revised manuscript (Lines 222-241).

Reviewer #3 (Remarks to the Author):

Zhang et al. are presenting results of an experimental study on heterogeneous SO₂ oxidation carried out on irradiated soot/black carbon that originates from a Diesel engine (DBC). Analysis was conducted by means of DRIFTS following the IR feature, which is attributed to sulfate formation. The authors try to follow from the observed sulfate signal that sulfuric acid is formed. Reactive intermediates were measured by spin-trapping and a reaction scheme of heterogeneous SO₂ oxidation on DBC is proposed.

This manuscript is an interesting and timely work and is well written. New pathways of SO₂ oxidation leading to sulfuric acid (SA) represent a hot topic in atmospheric sciences. Possible heterogeneous processes in this context are under debate since a couple of years.

Answer: Thank you very much for giving us these valuable comments. The detailed

revisions have been added into the revised manuscript according to your comments (See below).

Here are my comments:

Q1. (1) In the Abstract etc. it sounds like the authors mean that DBC-based SO₂ oxidation could lead to additional gas-phase SA in the atmosphere, right? Or do they mean only surface SA? Please clarify. (2) No experimental proof of gas-phase SA formation is presented for atmospheric conditions! (3) Moreover, DRIFTS only shows the formation of sulfate at the surface. The argumentation that it is likely SA at the surface (in part) is not convincing to me.

A1. (1) Based on the original DRIFTS results and TGA analysis, we can only confirm the existence of surface H₂SO₄. In fact, it is quite difficult to identify whether particulate H₂SO₄ is directly formed on the surface or from the condensation of gaseous H₂SO₄. ESR analysis results show that ROS are produced on the DBC surface under illumination. Therefore, it is considered that surface oxidation plays an important role in the formation of H₂SO₄. Nevertheless, it cannot yet exclude the formation of gaseous SA in this reaction.

(2) A just published paper by He et al. reported that soot surfaces upon irradiation can give rise to gaseous OH radical¹⁷. Thus, the gas-phase oxidation of SO₂ by gaseous OH radicals may produce gaseous H₂SO₄ and also contribute to particulate H₂SO₄ due to condensation. To verify the presence of gaseous H₂SO₄ in this reaction, the heterogeneous reaction of SO₂ (~60 ppb) on DBC (~0.25 g) powders was carried out in a quartz flow reactor (Fig. R9). Airflow passing the illuminated black carbon surfaces, which may contain SO₂ and gaseous H₂SO₄, further passed through a solution containing formaldehyde (4 ml, 11% v/v), which is used to exclude the interference of the direct aqueous oxidation of SO₂ to sulfate. As seen in Fig. R10, it could be found that the concentration of SO₄²⁻ ions (1.0±0.28 μg ml⁻¹) after 24 h reaction under light irradiation was greater than that (0.7±0.1 μg ml⁻¹) under dark conditions. This result provides direct evidence for the possible existence of gaseous sulfuric acid, especially

under light irradiation. Thus, the condensation of gaseous H_2SO_4 may also contribute to particulate H_2SO_4 .

This has been added into the revised manuscript (Lines 193-199).

Fig. R9. Schematic diagram of quartz photoreaction flow tank.

Fig. R10. The comparison of SO_4^{2-} concentrations under dark conditions and light irradiation

(3) DRIFTS results cannot accurately identify whether the product is in the form of sulfuric acid. XPS analysis showed that there was no metal in the sample. At the same time, the pH of the sample after the reaction decreased from 3.74 to 2.68, indicating the formation of H^+ on surface. Thus, there is a great possibility that sulfate exists in the form of sulfuric acid. Moreover, TGA-MS measurement results showed that the

thermo-desorption behavior of the product on DBC is very similar to that of directly mixed sulfuric acid, which further confirmed the presence of H₂SO₄ on the surface.

Q2. (1) The desorption/pyrolysis results in Fig. 1, panel D, are based on total traces of SO and SO₂ fragments, which are not specific for SA. At temperature higher than 350 °C, SA starts to decompose forming SO₃ and H₂O. How does it influence data analysis? (2) The TGA-MS experiments are not described in Methods.

A2. (1) Thank you for your suggestions, which make us rethink the role of thermogravimetry. We agree with you that both SO and SO₂ fragments are not specific to H₂SO₄. From Fig. R11, the fragment component of the H₂SO₄ mass spectrum is mainly composed of the ions at m/z 98 (H₂SO₄), 81 (HSO₃), 80 (SO₃), 64 (SO₂), and 48 (SO).¹⁸⁻²⁰ A new TGA-MS experimental result of ~60 μL pure H₂SO₄ (Fig.R12A) showed that the major weight loss happens in the temperature range of 200 °C to 400 °C. Thus, the evaporation and decomposition of H₂SO₄ take place simultaneously in this temperature range, which can be supported by the observation of both H₂O (m/z 18) and SO₃ (m/z 80) (Fig. R12B). Unlike the MS spectrum of H₂SO₄, m/z=80 is not the main fragment of SO₃. Meanwhile, the sulfur-containing fragments such as SO₂ (m/z 64) and SO (m/z 48) mainly originate from the further fragmentation of SO₃ during electronic ionization.²⁰ Here, both SO and SO₂ fragments are chosen to trace the formation of H₂SO₄ due to the high signal intensity in TGA-MS analysis.

This discussion has been added into the revised manuscript (Lines 176-190) and Supporting Information (Lines 71-82)

Fig. R11. The mass spectrum of H_2SO_4 (from NIST Chemistry Webbook)

Fig. R12. The evolution of other fragment ions with temperature during TGA-MS analysis of H₂SO₄

(2) The sulfur-containing species produced during diesel soot photooxidation were investigated using a combined method of thermogravimetry -mass spectrometry (TGA-MS). In brief, thermogravimetry (TGA, DSC1-1600HT, Mettler-Toledo) was coupled with quadrupole mass spectrometry (MS, ThermoStar- GSD/350, Pfeiffer Vacuum) by silica capillary at a temperature of 150 °C. The MS system was equipped with a 70-eV electron impact ionizer and provided the mass spectra up to m/z 300. TGA-MS was carried out over the range 35-800 °C with a 30 K min⁻¹ gradient. The whole experiment was accomplished under an inert purge gas of N₂ with a constant flow rate of 70 ml min⁻¹. In a typical desorption run, a blank test of the empty sample crucible was performed at 25 °C in a N₂ stream. Afterwards, the sample was weighed and placed in

the sample container. The pure DBC (15 mg), SO₂-aged DBC (15 mg), pure H₂SO₄ solution (60 μL), and a H₂SO₄-DBC mixture (30 μL H₂SO₄ (1.8 mol/L) and 15 mg DBC) were in sequence placed in an alumina crucible of 70 μL and then purged with N₂ for 1 hour. The mass spectra of m/z 98, 80, 64, and 48, which correspond to the main fragment components of H₂SO₄ (H₂SO₄, SO₃, SO₂, SO, respectively), were chosen to further verify the occurrence of H₂SO₄.

The introduction on TGA-MS experiments has been rewritten in the section on “Methods” (Line 410-424).

Q3. Yao et al., ES&T Lett. (2020), 10.1021/acs.estlett.0c00615, proposed SO₃ formation from soot-based SO₂ oxidation supported by direct SO₃ measurements in the atmosphere. Can the authors rule out a similar process here? Please discuss this topic.

A3. Thank you for your reminder, we have noted the work by Yao et al.²¹ Recently, Yao et al observed a good correlation between SO₃ formation and traffic-related soot during the early morning in Beijing.²¹ They further proposed that the surface catalytic oxidation of SO₂ on the ether group sites of soot surfaces may be the crucial contributor of SO₃ according to the DFT calculation results by He et al.²² Based on the IC results in Fig. R4B, it could be found that H₂SO₄ produced on SiO₂/DBC under dark conditions (2.0±0.10 μg ml⁻¹) was greater than that in the control experiments (0.35±0.02 μg ml⁻¹ for fresh SiO₂/DBC mixture). This indicated that the heterogeneous conversion of SO₂ on DBC can also promote H₂SO₄ formation under dark conditions. Here, we think that this catalytic process under dark conditions could also occur on DBC. Nevertheless, the production of sulfuric acid under light conditions is significantly greater than that under dark conditions (Fig.R4B), indicating that the oxidation of SO₂ by photoinduced radicals should be the main route of sulfuric acid formation in the photochemical reactions.

Fig.R4. SO₂ uptake on DBC under different conditions (A); comparison of H₂SO₄ concentrations under different conditions (B).

This has been added into the revised manuscript (Lines 140-144, and Line 318-321).

Q4. Line 123: Atmospheric SO₂ levels are a few ppbv. In the experiments, “low SO₂” was about 1 ppm, i.e. about a factor of 1000 higher. Is the heterogenous SO₂ conversion that slow that such high SO₂ is needed? What does it mean for the reaction rate for atmospheric conditions and its importance for global SO₂ oxidation?

A4. Generally, high-level SO₂ in *in-situ* DRIFTS experiments is necessary to easily observe the remarkable discrepancies among different experimental conditions and further explore the reaction mechanism. We agree with you that atmospheric SO₂ levels are a few ppb. In China, the average SO₂ concentration was in the range of 50 - 75 ppb from 2012 to 2014.³ In the last few years, the average SO₂ concentration has exhibited

a constant decrease due to the reduction in the emission of SO₂. Even so, the hourly average concentration of SO₂ was still up to ~12 ppb during the period of winter residential heating in Beijing (Fig. R3). Thus, to simulate the heterogeneous conversion of SO₂ on the DBC surface under close to real atmospheric conditions, the photooxidation experiments of low-level SO₂ (10 ppb and 60 ppb) on DBC were carried out in the flow reactor (Fig.R1). Fig. R4A shows the total uptake of 60 ppb SO₂ on particles under different conditions. When SiO₂ particles in a tube plug flow reactor were exposed to SO₂ (60 ppb), SO₂ concentrations decreased slightly and returned to the initial concentration in 8 min. This indicated that the reaction of SO₂ on the fresh SiO₂ surface is weak. In contrast, a sharp decrease in SO₂ concentration was detected in the DBC/SiO₂ system under either dark conditions or light irradiation. The uptake process lasted much more than 10 h. Under light irradiation, the total SO₂ uptake on DBC was slightly greater than that under dark conditions. These results indicate that the irreversible uptake of SO₂ on DBC is significant even under conditions close to the real atmosphere. To further quantitatively assess the heterogeneous conversion of SO₂ on DBC, extracted SO₄²⁻ ions were analyzed using IC (Fig. R4B). Clearly, the SO₄²⁻ concentration is the highest (3.7±0.59 μg ml⁻¹ for 60 ppb SO₂) in the presence of DBC under light irradiation compared with the control experiments. This further proved that both light irradiation and DBC have a significant enhancing role on the heterogeneous conversion of SO₂. Moreover, the higher SO₄²⁻ concentration obtained at 10 ppb SO₂ (1.5±0.76 μg ml⁻¹ for 10 ppb SO₂) further highlights the enhancing role of DBC photooxidation at lower SO₂ levels.

Fig. R3. The hourly average concentration of SO_2 from 30 July 2020 to 20 May 2021 in Beijing.

Fig. R1. Schematic diagram of the tube plug flow reactor setup.

Fig.R4. SO₂ uptake on DBC under different conditions (A); comparison of H₂SO₄ concentrations under different conditions (B).

Fig. R2. SO₂ uptake on DBC with different mass concentrations (A); H₂SO₄ formation rate as a function of DBC concentration (B).

To quantitatively evaluate the role of DBC in H₂SO₄ formation rates under conditions close to the real atmosphere, photooxidation experiments of SO₂ (~60 ppb) on DBC of different mass concentrations were carried out in a flow reactor. The schematic diagram of experimental setup is shown in Fig. R1. It was found that SO₂ uptake increases apparently with DBC mass concentration (Fig. R2A). The measured H₂SO₄ formation rates varied linearly with the mass concentration of DBC under light irradiation (Fig. R2B) and the mass normalization rate is determined to be $\sim 1.0 \times 10^{-3} \text{ h}^{-1}$. BC mass concentrations have been observed to range from 10.4 to 17.8 $\mu\text{g m}^{-3}$ during haze events.¹ On this basis, the H₂SO₄ formation rates in the real atmosphere were estimated to be in the range of 0.01-0.018 $\mu\text{g m}^{-3} \text{ h}^{-1}$ according to the linear equation obtained in this work. This was comparable to the formation rate of gaseous H₂SO₄ (~ 0.001 - 0.1 $\mu\text{g m}^{-3} \text{ h}^{-1}$) from the OH reaction pathway under relatively clean conditions.² These results suggest that the photooxidation of SO₂ on the surface of black carbon could be an important source of sulfate in areas with high black carbon loading.

This has been added into the revised manuscript (Lines 133-158).

Q5. Line 177, spin-trapping: It is nice to have these results. O_2^- and OH are the reactive intermediates, well in line with that what can be expected from knowledge in literature. Or can we draw any special conclusion for this system here?

A5. An important special conclusion on the formation of OH radicals can be drawn from this work. The most recent work published in *Angew. Chem.* (2022, 134, e2022016) reported that the reaction between water and O_2 on carbonaceous soot surfaces under light irradiation can give rise to the formation of gaseous OH radicals via triggering the formation and conversion of singlet O_2 (1O_2).¹⁷ In this study, our experimental results proved that the transmission of photo-induced electrons and the conversion of superoxide ion (O_2^-) on the DBC surface could also contribute to the formation of OH radicals. Thus, our work further complements or improves the production mechanism of OH radicals on DBC under light irradiation and further confirmed its environmental significance.

These discussions have been added into the revised manuscript (Lines 307-314).

Q6. All in all, it is a nice piece of work in the field of heterogeneous SO_2 oxidation. But I see not special novelty value or it is not clearly communicated. If the authors mean that the described process is important for atmospheric SA production, gas-phase SA measurements for close to atmospheric conditions should be added to verify this hypothesis. The manuscript in its present form is more suited for a special journal dealing with heterogeneous catalysis in the atmosphere or atmospheric chemistry generally.

A6. We appreciate your affirmation of our work. In fact, inspired by the valuable comments and suggestions of all reviewers, we reconsidered the significance of this research. We think this research has implications for the wider scientific community in the following aspects.

Firstly, we verified the formation of gas-phase SA at close to atmospheric conditions. As mentioned in A1(2), this study provides direct evidence for the possible existence of gaseous sulfuric acid, especially under light irradiation. This is quite meaningful for further understanding the formation and growth of new particle in the atmosphere.

This discussion has been added into the revised manuscript (Lines 315-327).

Secondly, the conventional view is that BC particles mainly act as a reducing agent. For example, BC is considered to play an important role in the heterogeneous reduction of NO₂ to HONO in the atmosphere.^{1,23} In contrast, in this work, we first reported that DBC can act as an oxidation medium to directly promote the heterogeneous oxidation of SO₂ to H₂SO₄ by OH radicals. The current three-dimensional air quality models usually underestimate the concentrations of sulfate due to unknown formation pathways.²⁴⁻²⁶ Thus, our work may provide new insight into the long-standing puzzle regarding an unidentified surface oxidation channel of SO₂.

This discussion has been added into the revised manuscript (Lines 300-305).

Thirdly, the photochemical aging of SO₂ could change the optical properties and climate effect of BC. It is well accepted that black carbon has strong effects on regional and global climates due to the remarkable positive (warming) radiative forcing.²⁷⁻²⁹ The change of mixing state caused by aging in the atmosphere is an important reason for the uncertainty in the prediction of the radiative forcing of black carbon.³⁰⁻³² This study confirmed the possibility of mixing black carbon and sulfuric acid in the atmosphere. Thus, future study on the optical properties of BC internally mixed with sulfuric acid is necessary for better evaluating the climate effects of aged BC. Moreover, mixing with acidic sulfuric acid will also greatly increase the health risk of BC.

This discussion has been added into the revised manuscript (Lines 331-339).

References

1. Zhang, F.; Wang, Y.; Peng, J. F.; Chen, L.; Sun, Y. L.; Duan, L.; Ge, X. L.; Li, Y. X.; Zhao, J. Y.; Liu, C.; Zhang, X. C.; Zhang, G.; Pan, Y. P.; Wang, Y. S.; Zhang, A. L.; Ji, Y. M.; Wang, G. H.; Hu, M.; Molina, M. J.; Zhang, R. Y., An unexpected catalyst dominates formation and radiative forcing of

- regional haze. *Proc. Natl. Acad. Sci. U. S. A.* **2020**, *117*, (8), 3960-3966.
2. Cheng, Y. F.; Zheng, G. J.; Wei, C.; Mu, Q.; Zheng, B.; Wang, Z. B.; Gao, M.; Zhang, Q.; He, K. B.; Carmichael, G.; Poschl, U.; Su, H., Reactive nitrogen chemistry in aerosol water as a source of sulfate during haze events in China. *Science Advances* **2016**, *2*, (12).
 3. Zhang, F.; Wang, Y.; Peng, J.; Chen, L.; Sun, Y.; Duan, L.; Ge, X.; Li, Y.; Zhao, J.; Liu, C., An unexpected catalyst dominates formation and radiative forcing of regional haze. *Proceedings of the National Academy of Sciences* **2020**, *117*, (8), 3960-3966.
 4. Tsona, N. T.; Du, L., A potential source of atmospheric sulfate from O₂-induced SO₂ oxidation by ozone. *Atmospheric Chemistry and Physics* **2019**, *19*, (1), 649-661.
 5. Tsona, N. T.; Li, J.; Du, L., From O₂-Initiated SO₂ Oxidation to Sulfate Formation in the Gas Phase. *The Journal of Physical Chemistry A* **2018**, *122*, (27), 5781-5788.
 6. Radi, R., Oxygen radicals, nitric oxide, and peroxyxynitrite: Redox pathways in molecular medicine. *Proc. Natl. Acad. Sci. U. S. A.* **2018**, *115*, (23), 5839-5848.
 7. Wang, Y. B.; Zhao, X.; Cao, D.; Wang, Y.; Zhu, Y. F., Peroxymonosulfate enhanced visible light photocatalytic degradation bisphenol A by single-atom dispersed Ag mesoporous g-C₃N₄ hybrid. *Applied Catalysis B-Environmental* **2017**, *211*, 79-88.
 8. Gao, H. Y.; Huang, C. H.; Mao, L.; Shao, B.; Shao, J.; Yan, Z. Y.; Tang, M.; Zhu, B. Z., First Direct and Unequivocal Electron Spin Resonance Spin-Trapping Evidence for pH-Dependent Production of Hydroxyl Radicals from Sulfate Radicals. *Environ. Sci. Technol.* **2020**, *54*, (21), 14046-14056.
 9. Zhu, J. L.; Shang, J.; Zhu, T., A new understanding of the microstructure of soot particles: The reduced graphene oxide-like skeleton and its visible-light driven formation of reactive oxygen species. *Environ. Pollut.* **2021**, 270.
 10. Zhao, Y. C.; Liu, Y.; Zhang, X. B.; Liao, W. C., Environmental transformation of graphene oxide in the aquatic environment. *Chemosphere* **2021**, 262.
 11. Hou, W. C.; Chowdhury, I.; Goodwin, D. G.; Henderson, W. M.; Fairbrother, D. H.; Bouchard, D.; Zepp, R. G., Photochemical Transformation of Graphene Oxide in Sunlight. *Environ. Sci. Technol.* **2015**, *49*, (6), 3435-3443.
 12. Yuan, Y. C.; Jin, N.; Saghy, P.; Dube, L.; Zhu, H.; Chen, O., Quantum Dot Photocatalysts for Organic Transformations. *J. Phys. Chem. Lett.* **2021**, *12*, (30), 7180-7193.
 13. Kampouri, S.; Stylianou, K. C., Dual-Functional Photocatalysis for Simultaneous Hydrogen Production and Oxidation of Organic Substances. *Acs Catalysis* **2019**, *9*, (5), 4247-4270.
 14. Liu, W.; Li, Y. Y.; Liu, F. Y.; Jiang, W.; Zhang, D. D.; Liang, J. L., Visible-light-driven photocatalytic degradation of diclofenac by carbon quantum dots modified porous g-C₃N₄: Mechanisms, degradation pathway and DFT calculation. *Water Res.* **2019**, *150*, 431-441.
 15. Song, H.; Meng, X. G.; Wang, S. Y.; Zhou, W.; Song, S.; Kako, T.; Ye, J. H., Selective Photo-oxidation of Methane to Methanol with Oxygen over Dual-Cocatalyst-Modified Titanium Dioxide. *Acs Catalysis* **2020**, *10*, (23), 14318-14326.
 16. Li, M.; Bao, F. X.; Zhang, Y.; Song, W. J.; Chen, C. C.; Zhao, J. C., Role of elemental carbon in the photochemical aging of soot. *Proc. Natl. Acad. Sci. U. S. A.* **2018**, *115*, (30), 7717-7722.
 17. He, G.; Ma, J.; Chu, B.; Hu, R.; Li, H.; Gao, M.; Liu, Y.; Wang, Y.; Ma, Q.; Xie, P., Generation and Release of OH Radicals from the Reaction of H₂O with O₂ over Soot. *Angew. Chem.* **2022**.
 18. Drewnick, F.; Diesch, J. M.; Faber, P.; Borrmann, S., Aerosol mass spectrometry: particle-vaporizer interactions and their consequences for the measurements. *Atmospheric Measurement Techniques* **2015**, *8*, (9), 3811-3830.

19. Jimenez, J. L.; Jayne, J. T.; Shi, Q.; Kolb, C. E.; Worsnop, D. R.; Yourshaw, I.; Seinfeld, J. H.; Flagan, R. C.; Zhang, X. F.; Smith, K. A.; Morris, J. W.; Davidovits, P., Ambient aerosol sampling using the Aerodyne Aerosol Mass Spectrometer. *Journal of Geophysical Research-Atmospheres* **2003**, *108*, (D7).
20. Smith, O. I.; Stevenson, J. S., Determination of Cross-Sections for Formation of Parent and Fragment Ions by Electron-Impact from SO_2 and SO_3 . *J. Chem. Phys.* **1981**, *74*, (12), 6777-6783.
21. Yao, L.; Garmash, O.; Bianchi, F.; Zheng, J.; Yan, C.; Kontkanen, J.; Junninen, H.; Mazon, S. B.; Ehn, M.; Paasonen, P., Atmospheric new particle formation from sulfuric acid and amines in a Chinese megacity. *Science* **2018**, *361*, (6399), 278-281.
22. He, G. Z.; Ma, J. Z.; He, H., Role of Carbonaceous Aerosols in Catalyzing Sulfate Formation. *Acs Catalysis* **2018**, *8*, (5), 3825-3832.
23. Ammann, M.; Kalberer, M.; Jost, D.; Tobler, L.; Rössler, E.; Pignatelli, D.; Gäggeler, H.; Baltensperger, U., Heterogeneous production of nitrous acid on soot in polluted air masses. *Nature* **1998**, *395*, (6698), 157-160.
24. Zheng, H. T.; Song, S. J.; Sarwar, G.; Gen, M. S.; Wang, S. X.; Ding, D.; Chang, X.; Zhang, S. P.; Xing, J.; Sun, Y. L.; Ji, D. S.; Chan, C. K.; Gao, J.; McElroy, M. B., Contribution of Particulate Nitrate Photolysis to Heterogeneous Sulfate Formation for Winter Haze in China. *Environmental Science & Technology Letters* **2020**, *7*, (9), 632-638.
25. Fu, X.; Wang, S. X.; Chang, X.; Cai, S. Y.; Xing, J.; Hao, J. M., Modeling analysis of secondary inorganic aerosols over China: pollution characteristics, and meteorological and dust impacts. *Sci. Rep.* **2016**, *6*.
26. Xing, J.; Mathur, R.; Pleim, J.; Hogrefe, C.; Gan, C.-M.; Wong, D.-C.; Wei, C.; Gilliam, R.; Pouliot, G., Observations and modeling of air quality trends over 1990–2010 across the Northern Hemisphere: China, the United States and Europe. *Atmospheric Chemistry and Physics* **2015**, *15*, (5), 2723-2747.
27. Cappa, C. D.; Onasch, T. B.; Massoli, P.; Worsnop, D. R.; Bates, T. S.; Cross, E. S.; Davidovits, P.; Hakala, J.; Hayden, K. L.; Jobson, B. T.; Kolesar, K. R.; Lack, D. A.; Lerner, B. M.; Li, S. M.; Mellon, D.; Nuaaman, I.; Olfert, J. S.; Petaja, T.; Quinn, P. K.; Song, C.; Subramanian, R.; Williams, E. J.; Zaveri, R. A., Radiative Absorption Enhancements Due to the Mixing State of Atmospheric Black Carbon. *Science* **2012**, *337*, (6098), 1078-1081.
28. Jacobson, M. Z., Strong radiative heating due to the mixing state of black carbon in atmospheric aerosols. *Nature* **2001**, *409*, (6821), 695-697.
29. Gustafsson, O.; Krusa, M.; Zencak, Z.; Sheesley, R. J.; Granat, L.; Engstrom, E.; Praveen, P. S.; Rao, P. S. P.; Leck, C.; Rodhe, H., Brown Clouds over South Asia: Biomass or Fossil Fuel Combustion? *Science* **2009**, *323*, (5913), 495-498.
30. Zhang, R. Y.; Khalizov, A. F.; Pagels, J.; Zhang, D.; Xue, H. X.; McMurry, P. H., Variability in morphology, hygroscopicity, and optical properties of soot aerosols during atmospheric processing. *Proc. Natl. Acad. Sci. U. S. A.* **2008**, *105*, (30), 10291-10296.
31. Peng, J. F.; Hu, M.; Guo, S.; Du, Z. F.; Zheng, J.; Shang, D. J.; Zamora, M. L.; Zeng, L. M.; Shao, M.; Wu, Y. S.; Zheng, J.; Wang, Y.; Glen, C. R.; Collins, D. R.; Molina, M. J.; Zhang, R. Y., Markedly enhanced absorption and direct radiative forcing of black carbon under polluted urban environments. *Proc. Natl. Acad. Sci. U. S. A.* **2016**, *113*, (16), 4266-4271.
32. Liu, D. T.; Whitehead, J.; Alfarra, M. R.; Reyes-Villegas, E.; Spracklen, D. V.; Reddington, C. L.; Kong, S. F.; Williams, P. I.; Ting, Y. C.; Haslett, S.; Taylor, J. W.; Flynn, M. J.; Morgan, W. T.; McFiggans, G.; Coe, H.; Allan, J. D., Black-carbon absorption enhancement in the atmosphere determined by particle

mixing state. *Nature Geoscience* **2017**, *10*, (3), 184-U132.

REVIEWERS' COMMENTS

Reviewer #1 (Remarks to the Author):

The MS has been improved, and all my comments have been taken into account.

The MS can be published as it is.

Very good MS and important contribution.

Reviewer #2 (Remarks to the Author):

This revised version has greatly improved compared to the initial submission. Even if the atmospheric implication is not fully convincing, I would recommend its publication in its present form as it highlights that SO₂ may undergo heterogeneous chemistry that is not fully understood. Going beyond such as statement is certainly not supported by the reported data.

Reviewer #3 (Remarks to the Author):

Ms. No. NCOMMS-22-11710A

I've read the revised version of the manuscript and the response to reviewer's comments. The authors addressed all the issues mentioned.

Now, the authors also added a paragraph dealing with the (potential) experimental verification of gas phase sulfuric acid production. That's very good. Unfortunately, an estimate regarding the importance of this new chemistry for the sulfuric acid balance is still missing.

Nevertheless, I'm now thinking that the manuscript in its present form could be very interesting for the readership from different disciplines.

Thus, acceptance can be recommended.

Point-by-Point Response to the Reviewers' Comments

Responses to Comments from Reviewer #1:

The MS has been improved, and all my comments have been taken into account. The MS can be published as it is. Very good MS and important contribution.

Answer: Thank you so much again for your positive comments.

Responses to Comments from Reviewer #2:

This revised version has greatly improved compared to the initial submission. Even if the atmospheric implication is not fully convincing, I would recommend its publication in its present form as it highlights that SO₂ may undergo heterogeneous chemistry that is not fully understood. Going beyond such a statement is certainly not supported by the reported data.

Answer: Thank you so much again for your positive comments.

Responses to Comments from Reviewer #3:

I've read the revised version of the manuscript and the response to reviewer's comments. The authors addressed all the issues mentioned. Now, the authors also added a paragraph dealing with the (potential) experimental verification of gas phase sulfuric acid production. That's very good. Unfortunately, an estimate regarding the importance of this new chemistry for the sulfuric acid balance is still missing. Nevertheless, I'm now thinking that the manuscript in its present form could be very interesting for the readership from different disciplines. Thus, acceptance can be recommended.

Answer: Thank you so much again for your positive comments. We also thank for your valuable suggestion on evaluating the importance of SO₂ photooxidation mechanism on DBC. To quantitatively evaluate the significance of this new pathway to H₂SO₄ balance, further model simulation and field observation should be effectively combined in future studies, which is also the focus of our future work. We hope to resolve this problem through working with other model simulation teams in the future.

This has been added in the revised manuscript (Lines 340-343 in MS).

“It was worth noting that further model simulation and field observation in future studies should be effectively combined to quantitatively evaluate the contribution of this new pathway to H₂SO₄ formation in the atmosphere.”